# Plant-Based Ingredients Utilized as Fat Replacers and Natural Antimicrobial Agents in Beef Burgers

**DOI:** 10.3390/foods13203229

**Published:** 2024-10-11

**Authors:** Silvia Jane Lombardi, Gianfranco Pannella, Francesca Coppola, Franca Vergalito, Lucia Maiuro, Mariantonietta Succi, Elena Sorrentino, Patrizio Tremonte, Raffaele Coppola

**Affiliations:** 1Department of Agricultural, Environmental and Food Sciences (DiAAA), University of Molise, Via De Sanctis snc, 86100 Campobasso, Italy; 2Department of Science and Technology for Sustainable Development and One Health, Università Campus-Bio-Medico di Roma, Via Alvaro del Portillo 21, 00128 Rome, Italy; 3Institute of Food Science, National Research Council, Via Roma, 60, 83100 Avellino, Italy

**Keywords:** better meat, bio-preservation, albedo, medlar, *Eriobotrya japonica*, seed extract

## Abstract

The present study aimed to find solutions based on the use of plant-based ingredients that would improve the nutritional quality of meat products as well as ensure sensory and microbiological quality. Two fat replacers, lemon albedo (*Citrus lemon*) and carob seed gum (*Ceratonia siliqua*), were investigated by chemical analysis and panel testing to evaluate their effect on the nutritional and sensory quality of beef burgers. The antimicrobial activity of two plant extracts, from nettle (*Urtica dioica*) leaves and medlar (*Eriobotrya japonica*) seeds, was studied, evaluating the intensity of inhibitory action and the minimum inhibitory concentration against *Pseudomonas* spp. and *Listeria innocua* strains by plate test. In addition, the antioxidant activity of both extracts was evaluated. Based on the results, lemon albedo and medlar seed extracts were validated in a food model (beef burger) by a storage test and a challenge test. The storage test results highlight that medlar seed extract prevents the formation of thiobarbituric acid reactive substances (TBARSs) and ensures microbiological quality, inhibiting *Enterobacteriaceae* and *Pseudomonas* spp. Anti-*Listeria* efficacy was confirmed in situ by challenge test results. In conclusion, although fat replacers ensure nutritional and sensory quality, they do not satisfy microbiological quality. This study clearly demonstrates that the safety of low-fat burgers can only be achieved through the combination of appropriate fat replacers with well-selected natural antimicrobial extracts.

## 1. Introduction

The use of new plant-based ingredients that can simultaneously improve the nutritional quality and safety of meat products is the most important challenge facing the meat industry.

Meat and meat products are the most controversial food products due to their disadvantageous position in terms of environmental sustainability, challenging climate change, and promoting health and safety [1,2,3,4].

Recently, Kennedy et al. [2] showed that a reduction in processed meat and animal fat consumption could lead to substantial health co-benefits by reducing the incidence of type 2 diabetes, cardiovascular disease, and colorectal cancer. “Less but better meat” is a pragmatic approach that has become increasingly popular in recent years to address the challenges of sustainable meat consumption and production [5]. The strategies to achieve better meat are diverse; a reduction in fats or their replacement by alternative ingredients (plant-based fat substitute or plant-based fat mimetic), a reduction in synthetic additives (using natural antimicrobial agents), and increasingly less invasive ways of preparation and consumption are just some of the issues that are driving the challenge for a better meat [6,7]. In recent decades, much attention has been paid to identifying and experimenting with fat substitutes for use in the preparation of lower-calorie meat products. Fiber-rich plant sources, hydrocolloids, and animal-protein-rich matrices have shown promising results in replacing the added lipid fraction [8,9,10,11,12]. However, several fat substitutes have both advantages and limitations in their use [10]. Due to the importance of fat in defining the taste and rheological properties of meat products, fat replacement certainly has a strong impact on sensory and rheological properties. According to the literature, the use of fiber-based fat substitutes best protects the sensory and technological characteristics of meat products [13]. Several authors have found that the use of vegetable fibers in chicken patties and beef burgers improves cooking performance and rheological characters [14,15], effects that may be attributable to the cross-linking ability of vegetable fibers with meat proteins.

However, to date, although many fat substitutes are available on the market, there are still many challenges to be overcome to improve their sensory perception [16]. Authors have recently reported that the reduction and replacement of fat in meat products leads to changes in physiological and sensory properties. They emphasize that unresolved questions remain concerning texture, hardness, oxidative stability, juiciness, viscosity and overall acceptability. More specifically, the authors of the recent review [16] point out that each macro-typology of fat substitutes shows substantial limitations on the sensory level. Whey protein-based substitutes have negative effects on hardness, chew-ability and adhesiveness; carbohydrate-based substitutes (starch or inulin) improve hardness but result in disadvantages such as graininess and burnt flavor after cooking; and finally, lipid-based substitutes (oils from vegetable sources) are characterized by oxidation problems and, in some cases, gumminess. In addition, the use of fat replacers as alternative ingredients does not always guarantee microbiological quality and safety. In this regard, it should be considered that the microbiological quality of fresh meat is known to be a critical concern since its intrinsic characteristics allow the growth of various spoilage and pathogenic microorganisms [17,18]. The addition of fat replacers, which further increase the moisture content of the meat product [19,20], can further affect the microbiological quality [21]. The latter authors reported that the use of albedo in fermented meat products caused an increase in moisture and water activity (a_w_). Consequently, the use of fat replacers should not be separated from the use of antimicrobial preservatives. However, the use of conventional preservatives would conflict with clean-label qualification requirements that encourage a reduction in additive use. However, the use of natural additives would satisfy the two opposing requirements (microbiological safety and clean label). The antimicrobial activity of plant extracts and their protective action in food systems have been extensively studied in recent decades [22,23,24,25]. Furthermore, their action against spoilers and pathogens in meat products has been investigated [26,27]. Interesting antimicrobial activities have been observed in the extracts of a myriad of plants, ranging from the most popular to those of specific geographical areas [28,29,30]. Extracts from the leaves of the widest range of plants are among the best-studied antimicrobial agents [31,32]. *Urtica dioica* (nettle), one of the most widespread wild herbs in the world (native to Europe, Asia, North America and North Africa), is used for extract production from its leaves, whose antimicrobial properties have been studied in recent years [33]. Less investigated but equally of interest are extracts from seeds [34,35]. Although there is plenty of evidence of protection for extracts of leaves and, in some cases, also of plant seeds, to our knowledge, there is a lack of in vitro or food-model studies on the combined use of antimicrobial plant extracts and fat substitutes.

Based on the above considerations, beef burgers were taken as a food model for which strategies to ensure the qualification of a better meat with high microbiological safety and nutritional quality were explored. Specifically, for the first time, the combination of fat replacers with extracts with high anti-*Listeria* and anti-*Pseudomonas* efficacy was tested. In this regard, the present study preliminarily selected the fat replacer, between two fiber matrices, and the antimicrobial plant extract, examining a leaf extract from nettle and a seed extract from *Eriobotrya japonica* (medlar). Fat substitutes were chosen based on sensory investigations as it is widely recognized that sensory quality assessment is not only innovation feedback [36,37] but also a useful tool for designing novel foods and ensuring their commercial success [38]. The vegetal extract with protective activity was selected on the basis of its antioxidant activity and its anti-*Pseudomonas* and anti-*Listeria* capacity. To better understand the specific action of the extract against the microorganisms in question, the microbial decay in the presence of the extracts was also studied through the application of microbiology predictive models aimed at ascertaining possible adaptations or resistance of the strains to the protective extracts. Finally, the selected antimicrobial extract and fat replacer were validated in the food model by storage and challenge tests.

## 2. Materials and Methods

### 2.1. Fat Replacer

Two plant-based sources, chosen for their sustainability and high fiber content, were evaluated for use as potential fat substitutes in the preparation of fresh meat products:

Albedo was obtained by freeze-drying the raw albedo portion, a by-product of lemon (*Citrus limon*) processing, as reported by Tremonte et al. [21]. Specifically, the raw material was treated at 90 °C for 5 min to ensure microbiological safety, frozen at −30 °C, and freeze-dried. To obtain a powder with a particle size of less than 0.417 mm, an appropriate grinder and sieves were used.

Carob bean gum (CBG), purchased from manufacturers (CioKarrua Ltd., Ragusa, Italy.) in Southern Italy, was obtained, as reported by the manufacturer, from hulled carob seeds, sieved, and ground to obtain native carob seed gum, which was then extracted with distilled water (90 °C for 60 min) and precipitated in isopropanol. The white fibrous precipitate formed was collected by filtration, dried under vacuum, and ground into a fine powder.

### 2.2. Selection of Fat Replacer

To choose the fat substitute, the suitability of freeze-dried albedo and carob bean gum in the preparation of hamburgers was evaluated. The same basic formulation consisting of lean beef, sodium chloride (2%), and ascorbic acid (0.1%) was used for all batches.

Specifically, the lean meat from the brisket cut, suitably defatted and deprived of visible connective tissue, was minced, and sodium chloride and ascorbic acid were added. The basic formulation after mixing was divided into three aliquots corresponding to the three batches prepared as described below:

C comprised burgers prepared with the mixture of salted and minced lean meat (90%) and beef fat (10%). Specifically, the fat was taken from non-low-melt cuts and shredded. The batch was used as a control.

A comprised burgers prepared with the mixture of salted and shredded lean meat (90%) and albedo (10%). Freeze-dried albedo was dissolved in water (1:2 *w*:*v*), stirred at 80 °C for 5 min, and cooled to a gelatinous consistency at 4 °C.

CBG comprised burgers prepared from the mixture of salted and shredded lean meat (90%) and carob bean gum (10%). The carob bean gum was suspended in water (1:2 *w*:*v*), stirred at 80 °C for 5 min, and cooled to a gelatinous consistency at 4 °C.

#### 2.2.1. Sensory Analysis

Sensory analysis was performed on the three batches of burgers to assess the following attributes: appearance, flavor, juiciness, residue, taste, tenderness, and visible fat. The sensory evaluation was conducted by 15 panelists who were not officially qualified (i.e., not registered with official organizations) but recruited among food technology experts (researchers and PhD students of the Department of Agricultural, Environmental and Food Sciences—University of Molise, Campobasso, Italy) and specifically trained in the sensory analysis of meat products with particular reference to burgers. Specifically, the training period, starting with 45 potential judges, continued for 5 weeks with two weekly sessions, until a group of 15 judges with a satisfactory uniformity in response was obtained. The panel of judges met in three sessions and in each session evaluated three burgers each for one batch. Thus, sensory analyses were performed in triplicate. The cooking procedure was performed in a cooker according to the method described by the American Meat Science Association methodology (AMSA, 2015) [39]. Cooking was conducted until either a final internal temperature of 72 °C was reached or recorded at the geometric center of each burger using a hypodermic probe thermocouple (model HYPOK60 ITSensor, Rovigo—Italy).

A tasting sheet was created to measure the intensity of each chosen attribute, using the 9-point hedonic rating scale, from 9 (“extremely pleasant”) to 1 (“extremely unpleasant”). Batches of hamburgers were identified with random 3-digit numbers and placed on plastic plates before serving. Three replicates of samples of each were sensory-analyzed.

#### 2.2.2. Chemical Composition of Low-Fat Hamburgers

Moisture, fat, protein, carbohydrate, fiber, and ash contents were determined in burgers from all batches according to the official methods of AOAC 2023 [40], specified below. In detail, the moisture of the burgers was determined based on 5 g of sample placed in an oven for 16 h at 105 °C, calculated as the difference between the weight of the fresh sample and the weight of the sample after drying and expressed as percentage of dry matter (d.m) [40]. Protein content was determined using the Kjeldhal method [40]. Crude fat content was measured in accordance with AOAC method 960.39 using a Soxhlet apparatus [40]. Ash content was determined by heating the samples for 3 h at 550 °C. Dietary fiber content was determined using AOAC method 985.29 [40]; carbohydrates (C) were calculated by the difference, as follows:C%=100−(M+P+L+A+F) 
where M = moisture (%); P = protein content (%); L = lipid content (%); A = ash content (%); F = dietary fiber (%).

#### 2.2.3. Microbiological Analysis

Undesirable microbial groups were enumerated as reported in the literature [21,22,23,24,25,26,27,28,29,30,31,32,33,34,35,36,37,38,39,40,41]. Briefly, 10 g samples were decimal-diluted in a sterile solution of 0.1% peptone water, homogenized in a Stomacher 400 Lab-blender (Seward Medical, London, UK) for 3 min, and serially diluted in the same sterile solution. Total mesophilic bacteria (TMC), *Pseudomonas* spp., *Enterobacteriaceae,* and *Listeria* spp. were detected after appropriate incubation in proper media and conditions as described in a previous study [42,43]. Briefly, TMC were enumerated on Plate Count Agar (PCA) (Oxoid, Milan, Italy) after incubation at 30 °C for 72 h. *Enterobacteriaceae* were enumerated on Violet Red Bile Glucose Agar (VRBGA) (Oxoid) after incubation at 37 °C for 24 h. *Pseudomonas* spp. were enumerated on *Pseudomonas* Agar Base (PAB) (Oxoid) supplemented with CFC *Pseudomonas* selective supplement (Oxoid) after 48 h of incubation at 25 °C, and *Listeria* spp. were enumerated on Listeria Selective Oxford Agar Base with Modified Listeria Selective Supplement (Oxoid, Milan, Italy) after 24 and 48 h of incubation at 37 °C. Each experiment was carried out in duplicate, and results were expressed as the mean of measurements and standard error.

### 2.3. Selection of Natural Extract

Natural extracts from nettle (*Urtica dioica*) leaves and medlar (*Eriobotrya japonica*) seeds were evaluated for their antimicrobial and antioxidant activity.

Extracts from the raw material were prepared in the laboratory as follows: nettle leaves, previously washed and dried, were desiccated in an oven at 40 °C for 48 h, while medlar seeds were used without being dried. The samples were finely ground and sieved using sieves with a diameter of 0.7 mm. Twenty-five grams of sample was resuspended in 300 mL of methanol and kept stirring at room temperature for 24 h. The solvent was removed by evaporation under reduced pressure at 40 °C. Each extract was suspended in the dimethyl sulfoxide (DMSO) solution.

#### 2.3.1. Antimicrobial Activity and Minimum Inhibitory Concentration Evaluation

The two extracts were evaluated for their antimicrobial activity against *Pseudomonas putida* DSMZ 291^T^, *Pseudomonas fluorescens* DSMZ 50009^T^, *Pseudomonas fragi* DSMZ 3456^T^ and *Listeria innocua* ATCC 33090 (DSMZ-German Collection of Microorganisms and Cell Cultures, Braunschweig, Germany). Strains of *Pseudomonas* and *L. innocua* were revitalized at 28 °C in Nutrient Broth (NB—Oxoid, Milan, Italy) or in Brain Heart Infusion (BHI—Oxoid, Milan, Italy), respectively. The inhibitory action of the extract was assessed by the agar well diffusion assay as previously described [41]. Briefly, 1 mL (inoculum concentration of 4.0 log CFU/mL) of each bacterial suspension was inoculated into 20 mL of proper soft media (0.7% agar), gently mixed, and poured into Petri plates (BHI for *L. innocua* and NB for *Pseudomonas* spp.) In each plate, 6 mm diameter wells were set up, within which 70 µL of nettle extract or medlar extract, previously solubilized in DMSO, was added. Tetracycline disk (30 µg) and DMSO (70 µL) were used as negative and positive control, respectively. As a control, wells containing only 70 µL of DMSO were set up. Plates inoculated with *Pseudomonas* spp. were incubated at 28 °C for 24 h, while plates containing *L. innocua* were incubated at 37 °C for 24 h. Antimicrobial activity was assessed by measuring the diameter of the halo formed around the wells after 24 h. Based on the halo diameter, the inhibitory activity was described as low (L; halo ≤ 1 cm), moderate (M; 1 < halo < 1.5 cm), or high (H; halo > 1.5 cm). The lack of halo formation was considered as absence of inhibition and indicated as “N”. Five independent replicates were performed.

Moreover, the minimum inhibitory concentration (MIC) was determined using the agar dilution method as described in EUCAST definitive document 3.1 [44] with some modifications. In detail, each extract was filter-sterilized and added to sterile molten Mueller–Hinton (MH) agar to produce a concentration range between 0.0002 and 0.4% (*w*/*v*). The resulting MH agar solutions were poured into 90 mm Petri plates and the surface of plates was inoculated with the microbial strains at a concentration of 4.0 log CFU/mL and incubated at 30 °C for 48 h. At the end of the incubation period, plates were evaluated for the presence or absence of growth. MIC values were recorded as the lowest concentration of natural extract that completely inhibited the growth. Each test was performed in triplicate.

The trend of the MIC values for *Pseudomonas* spp. and *Listeria innocua* in the presence of nettle leaf extract and medlar seed extract, both used 2 times, was also evaluated using the structured dynamic model of Baranyi and Roberts [45]. For this purpose, the DMFit (v2.0) software was used.

#### 2.3.2. Total Phenolic Content and Antioxidant Activity Evaluation

Polyphenol content of nettle and medlar extracts was determined using the Folin–Ciocalteu method [46]. A calibration line was constructed using gallic acid as a standard and the absorbance was read at 750 nm using a BioSpectrometer (Eppendorf, Hamburg, GE). The concentration of polyphenols was expressed as gallic acid equivalents (GAE mM).

The antioxidant activity was evaluated using the 2,2-DiPhenyl-1-PicrylHydrazyl (DPPH) radical scavenging method. Specifically, the DPPH assay was used, which allows the determination of the quantity of a compound that reduces the DPPH content by 50% (IC50); the absorbance was measured at 517 nm using a BioSpectrometer (Eppendorf, Hamburg, GE).

### 2.4. Albedo and Medlar Validation In Situ

The effect of albedo and medlar extract on the quality profile of the burgers was evaluated during cold storage at 4 °C for six days. To this end, a storage test and a challenge test were performed.

#### 2.4.1. Storage Test

To evaluate the effect of albedo and medlar extract on the shelf life of burgers, the samples were prepared as reported in Section 2.2 and divided into 4 aliquots corresponding to the 4 experimental batches, as detailed below:

C: Burgers containing 10% beef fat (control batch);

E: Burgers containing 10% fat fraction and medlar extract (0.5 mL/100 g burgers);

A: Burgers containing the albedo (10%) as fat substitute;

AE: Burgers containing 10% albedo (as fat replacer) and medlar extract (0.5 mL/100 g burgers).

The microbiological characteristics and oxidative damage during the storage period of six days were evaluated as follows:

The burgers were stored at 4 °C for six days and microbiological analyses were performed at time 0 and after 1, 2 3 and 6 days of storage. TMC were enumerated on PCA (Oxoid) incubated at 30 °C for 72 h and *Enterobacteriaceae* on VRBGA (Oxoid) after incubation at 37 °C for 24 h. In addition, oxidative damage was assessed during cold storage of hamburgers by malonyldialdehyde (MDA) determination. For this purpose, the thiobarbituric acid reactive substance (TBARS) test as described by Buege and Aust [47] was performed.

TBARSs were quantified to determine the degree of lipid oxidation, as described by Tremonte et al. [48], using trichloroacetic acid as solvent. Briefly, 10 g of hamburger homogenate was distilled, and an aliquot of the distillate was mixed with 2-thiobarbituric acid aqueous solution. After boiling, the absorbance of the chromofore formed was read by a spectrophotometer at 534 nm. TBARS values were expressed as μg of malonaldehyde per g (μg184 MDA/g) in dry matter.

#### 2.4.2. Validation of Anti-*Listeria* Effect

To simulate *Listeria* contamination during the process, a challenge test on hamburgers was carried out in parallel. For this purpose, the experimental design proposed by Sorrentino et al. [43] with some modifications was adopted. In detail, aliquots of each mixture batch described above were used and inoculated (1% *v*/*w*) with 4 log CFU/g of the multi-strain cocktail, a mixture of four strains of *L. innocua*: strain ATCC 33090 and strains Li03, Li12, and Li13 (belonging to the collection of Department of Agricultural, Environmental and Food Science, University of Molise). Prior to use, the microbial cultures were revitalized in BHI (Oxoid, Milan, Italy), and at the exponential phase, the strains were inoculated in the mixture for the challenge test, obtaining 4 experimental batches, CL, AL, EL, and AEL, which were prepared as the corresponding batches C, A, E and AE, but all inoculated with the multi-strain *Listeria* cocktail.

*Listeria* spp. trends in burgers stored at 4 °C were evaluated during the storage period (6 days). For this purpose, *Listeria* spp. were enumerated on *Listeria* Selective Oxford Agar Base with Modified *Listeria* Selective Supplement (Oxoid, Milan, Italy) after 24 and 48 h of incubation at 37 °C.

### 2.5. Statistical Analysis

Both data analysis and graph processing were carried out using the R (v4.2.3) environment and RStudio software (v2022.07.0). Data obtained from the sensorial analysis of hamburgers were analyzed to verify the hypothesis of the presence of significant differences (*p* < 0.05) between the burger samples prepared without fat replacer (Batch C), with albedo (Batch A), or with carob bean gum (Batch CBG) as fat replacers. In detail, the statistical test of Kruskal–Wallis, followed by Dunn’s post hoc test for multiple comparison, was used. To check the antimicrobial effect of the two extracts on the *L. innocua* and *Pseudomonas* spp., the analysis of variance (ANOVA) was conducted, followed by a Bonferroni post hoc test. Similarly, the ANOVA test with a Bonferroni post hoc test was used for the challenge test. The rstatix R package (0.7.2) [49] was used for this purpose.

## 3. Results and Discussion

### 3.1. Fat Replacer Selection

The development of the new hamburger involved the preliminary identification of the most suitable fat substitute through a sensory analysis. It is widely recognized that the evaluation of sensory quality is not only feedback of the innovations introduced [36,37] but is an increasingly useful tool for the design of new foods to ensure their commercial success [38]. Several studies report that plant-based dietary fibers used as fat replacements can improve the binding qualities and textural characteristics of meat products [50,51]. However, the choice of the appropriate fiber-based fat replacer does not disregard a specific evaluation of sensory analysis. In this regard, the effect of two dietary fiber-based fat replacers on burger sensory attributes was evaluated. Specifically, the effect produced by lemon albedo was compared with that expressed by carob seed flour.

Figure 1 shows the distribution of the level of approval of the sensory attributes of burgers prepared according to the conventional recipe, using fat in the control batch (C) or using albedo (A) or carob bean gum (CBG) as a fat substitute. Attention was paid to attributes perceptible not only by sight, such as general appearance and visible fat, but also by smell (e.g., flavor) and by palate, such as taste, juiciness, tenderness, and residue (attribute indicating the sense of permanence of the fat or substitute in the oral cavity). Both conventional burgers (C) and burgers prepared with the use of albedo (A) or carob bean gum (CBG) showed a good level of acceptability. However, in relation to the different attributes, sometimes significant differences (*p* < 0.05) emerged between the burger samples from the different batches. In particular, as reported in Appendix A, products prepared with albedo (A) showed better juiciness (median value 8), compared to both the control (C, median value 4) and CBG samples (median value 7). Moreover, the residue attribute (amount of residue remaining in the oral cavity after chewing) was associated with a better score for burgers with albedo (A, median value 3) compared to the control batch (C, median value 6) or the CBG batch (median value 7).

The sensory attributes of the samples obtained with carob bean gum (CBG) were comparable to those recorded for the control (C) and, in some cases, showed significantly lower levels of approval than those of the control (C). In particular, the use of carob bean gum negatively influenced the visible fat and residual attributes. In relation to the latter attribute, most of the panelists also described the perceived residue as sandiness. Therefore, the results call into question the use of carob bean gum in the development of the innovative burger. To date, the effect of carob bean gum on the sensory quality of processed meat products is quite controversial. Its use has been associated with positive effects by some authors [52,53], while other authors [54] have found undesirable sensory effects.

The different effects are probably due to the different quantities used: when the carob bean gum was utilized in concentrations close to 1%, no negative effects were found. In contrast, the negative sensory effect of carob bean gum found in our study could be traced back to the high concentrations used in the preparation of the hamburger. However, the decision to use carob bean gum at the 10% level is due to the ambition to develop burgers with a reduced-fat and fiber-source claim.

Albedo, in contrast to carob bean gum, also positively influenced sensory attributes when used at 10%, and consequently was chosen as a candidate for the most appropriate fat replacer.

Chemical results show that both albedo and carob bean gum used as a fat replacers significantly influenced the chemical and nutritional composition of the burgers, with a relevant increase in fiber and a reduction in fat content. However, since carob seed flour did not meet with good sensory acceptance, attention was turned to the effect produced by the use of albedo. Table 1, which shows the chemical parameters of the burgers obtained conventionally (Batch C) or with the addition of albedo (Batch A), shows how albedo enabled several claims to be achieved.

The fat replacement allows the content to be reduced from 36.9% to 6.9% with a consequent reduction in caloric content of more than 40%, a condition that satisfies the regulation (EC) [55] to qualify the product for the nutrition claim of reduced calories. Furthermore, the addition of albedo produces a significant contribution of fiber (14.25%/d.m. corresponding to 4.91 g per 100 g of product), allowing us to respond to the nutritional claim of it being a source of fiber.

Samples containing the albedo (batch A) as fat replacer are also characterized by significantly higher moisture levels than those of the control (batch C), a phenomenon attributable to the ability of albedo to bind and retain water. Similar results were also appreciated by other authors [56,57] who evaluated the possibility of using lemon fiber as a fat replacer. It is easy to see the technological and economic advantage of the increased water-holding capacity. This capacity, in fact, resulted in better cooking properties and a cooked product characterized by greater juiciness. Finally, the use of albedo involves an increase, albeit slight, in the level of simple carbohydrates. This last point, however, raises questions about the possible effect on the naturally contaminating microbiota. However, the higher moisture and carbohydrate content due to the addition of albedo raises questions about the possible effect on the naturally contaminating microbiota. Table 2 shows the main contaminants found in the hamburgers immediately after preparation. Samples from both batches showed significant levels of microbial load for most of the microbial groups analyzed, revealing no differences (*p* > 0.05) between the two batches. This finding highlights that albedo is not a source of potential microbial contamination. However, the levels of microbial load, in some respects not reassuring, that characterize both samples are worthy of attention and reflection. As widely demonstrated by several authors [21,48,58], the presence of such microorganisms could significantly compromise the shelf life and stability of meat products. Considering that burgers obtained by the addition of albedo are also characterized by slightly higher water activity (a_w_ 0.969) values than those of the control (a_w_ 0.961), the need for a protective strategy to safeguard microbiological safety and adequate shelf-life becomes evident. Therefore, in light of modern consumption trends where burgers are undercooked or even rare, the development of innovative bioprotection tools is of critical relevance.

### 3.2. Bio-Preservation Tool Development

To identify new and adequately performing plant extracts with preservative activity, the polyphenol content and antioxidant activity of two extracts obtained from nettle leaves (*Urtica dioica*) and medlar seeds (*Eriobotrya japonica*) were evaluated. The results reported in Table 3 highlight that both extracts are characterized by a polyphenolic content worthy of attention and comparable to or even higher than that found for other extracts previously investigated [22,58,59,60].

In detail, the nettle leaf extract is characterized by a phenol content of 125.64 mg GAE/g dry matter. Although it is not easy to compare this result with the scientific literature, the polyphenol content found in the present study for nettle extract is in accordance with the most flattering results. In fact, the level of phenols reported in the literature for nettle leaf extracts varies from a few mg GAE/g to slightly more than 100 mg GAE/g [61,62,63]. The phenol content shown by the extract medlar seeds is significantly higher (398.12 mg GAE/g dry matter). Such levels of phenols, as also reported in recent reviews [64,65], characterize the best-performing plant extracts. 

Since the phenolic content is known to be related to antioxidant and antimicrobial action [66,67,68,69], the two extracts examined in this work could represent an interesting bioprotective agent in food preservation. The antioxidant activity of the two extracts, assessed by means of the DPPH radical removal assay, showed a good level for both extracts. However, the activity was significantly (*p* < 0.05) better for the medlar seed extract than for the nettle leaf extract. The medlar extract with a level of 2.91 mg/mL resulted in a 50% reduction in the initial DPPH concentration, whereas in the case of the nettle extract, 21.53 mg/mL was required to achieve the same reduction.

Differences between the two extracts also remain in relation to antimicrobial activity. Figure 2 shows the antimicrobial action (expressed as a halo of inhibition) against *L. innocua* ATCC 33090 and against *Pseudomonas putida* DSMZ 291^T^, *Pseudomonas fluorescens* DSMZ 50009^T^ and *Pseudomonas fragi* DSMZ 3456^T^. Both extracts expressed good antimicrobial action against both the Gram-negative species *Pseudomonas* spp. and the Gram-positive *L. innocua*. The inhibiting action of nettle leaf extract against both Gram-positive and Gram-negative bacteria is already widely reported in the literature [70,71,72]. Medlar seed extract, for which no comprehensive information is available in the literature, exhibited pronounced and surprising antimicrobial activity. In particular, medlar extract produced an antimicrobial action significantly (*p* < 0.05) higher than that expressed by nettle extract and closer to that produced by the tetracycline disk, which showed an inhibition halo (Appendix A) of approximately 2.8 cm against all considered bacteria.

This finding is further supported by the MIC values: nettle leaf extract inhibited *Pseudomonas* species and *Listeria innocua* strains by displaying MIC values of 2 and 4 mg/mL (i.e., 0.2% and 0.4%), respectively, while medlar extract inhibited the three bacteria by exhibiting MIC values of 0.25 mg/mL (equivalent to 0.025%) in all cases. Such low MIC values for both *Pseudomonas* and *Listeria* make medlar extract an interesting bioprotection agent for meat products. *Pseudomonas* is a major contaminant and the most widespread meat spoilage agent [4,73]; consequently, its inhibition, especially by biotechnological means, is highly desirable.

In addition, *Listeria* inhibition is useful for product safety. Since *L. innocua* may be an interesting and accredited option as a pathogenic surrogate of *Listeria monocytogenes* [74], the evaluation of the extracts’ effect on its behavior is of crucial interest. Indeed, *Listeria innocua* not only bears a high similarity in genome sequence to the pathogen *L. monocytogenes* but has also proven to be a valid surrogate for the pathogenic species in specific thermal and non-thermal challenge tests [75,76].

Figure 3 shows the inactivation in terms of the MIC values of *L. innocua* in the presence of nettle leaf extract and medlar seed extract, both used 2 times. When nettle extract was used, the inactivation curve of *Listeria* was effectively described using the Baranyi and Roberts equation, showing a shoulder before inactivation and a tail at the end of the decay.

Specifically, after a shoulder of about 5 h, decay occurred at a rate of −0.17 h^−1^, and finally a tail was detected around a value of about 3.2 log CFU/mL.

The decay curve of *Listeria* in the presence of medlar seed extract is different. In this case, the trend is biphasic, characterized by a shoulder and decay with a rate approximately twice as fast (−0.31 h^−1^) as that observed in the presence of nettle extract. Furthermore, no tail was observed after the decay, *Listeria* having reached undetectable values already after 28 h. The ‘tail’ occurrence in the presence of nettle extract could be attributed to different factors, such as the presence of resistant cell subpopulations and adaptation phenomena to the antimicrobial treatment. For several bacteria, it has been highlighted that lethal or sub-lethal stress can induce resistance phenomena [77,78,79].

These data offer important information for the development of antimicrobial biotechnological agents. The absence of tails in the decay of *Listeria* caused by medlar extract suggests a lack of adaptation and resistance to the extract. These conditions are of great relevance considering that the identification of antimicrobial agents capable of not causing adaptation and resistance to *Listeria* is a serious need [80,81].

In the case of the strains referable to *Pseudomonas*, both extracts produced a decay characterized by an initial shoulder and a decay with a rate between −0.18 and −0.16 h^−1^ in the presence of the nettle leaf extract and between −0.31 and −0.35 h^−1^ in the presence of the medlar seed extract. In all cases, the charge levels reached inadmissible final values without showing any tailing phenomena. Based on the above evidence and considerations, medlar seed extract was selected as a biocontrol tool to be used in combination with fat replacer.

### 3.3. Fat Replacer and Bio-Preservation Tool Validation

The effectiveness of biotechnological tools has been evaluated and validated in situ through conservation and challenge tests. The multiplicity of ecological factors that characterize food could influence and reduce the activity expressed in vitro [82,83]. Just to mention a few of the complex phenomena that may occur in foods, polyphenols might interact through hydrogen and hydrophobic bonds with meat proteins and, as a result, be less bioavailable and bioactive [84]. For these and/or other interactions, a discrepancy between efficacies in vitro and in the food model is generally found [85]. Considering that, as also reported recently by Azevedo et al. [86], reproducibility in the food model of the efficacy demonstrated in vitro is a current challenge for the food industry; in the present study, the efficacy of the selected extract (used in combination with fat replacer) was validated in beef burgers using storage and challenge tests, as reported in the following subparagraphs.

#### 3.3.1. Storage Test

In our study, the effect of albedo (used as fat replacer) and medlar seed extract (used as preservative agent) on the microbiological and oxidative quality of burgers stored in a refrigerator for 6 days was evaluated. Enterobacteria and *Pseudomonas* trends as well as the total bacterial count during the storage period are reported in Figure 4.

For all the microbial groups, two different behaviors emerged depending on the mode of hamburger preparation. In particular, the samples of the control batches and those prepared with albedo showed an increasing trend over time without significant differences between them. Thus, while the albedo, as revealed in the previous paragraphs, satisfactorily meets nutritional and sensory requirements, it also confirms concerns about microbiological quality. It has already been reported in previous studies that albedo, increasing the moisture and carbohydrate content of meat products, favors microbial growth [21,22,23,24,25,26,27,28,29,30,31,32,33,34,35,36,37,38,39,40,41,42,43,44,45,46,47,48,49,50,51,52,53,54,55,56,57,58,59,60,61,62,63,64,65,66,67,68,69,70,71,72,73,74,75,76,77,78,79,80,81,82,83,84,85,86,87]. The results of our study showed that the samples prepared with albedo and the control samples were characterized by a noticeable increase in microbial populations. In particular, a marked increase in the levels of *Pseudomonas* spp. and *Listeria* was found from the second day of ripening, reaching alarming values at the end of storage. On the contrary, in the samples prepared with the use of medlar seed extract, alone or in combination with albedo, a substantial decrease in microbial loads during storage was detected. The antimicrobial action is therefore clearly attributable to the use of medlar seed extract. Other authors have already reported that plant extracts with a high phenolic substance content are able to exert an important protective action [88,89]. Therefore, data highlighted that the use of medlar extract in combination with albedo reassures against any concerns about microbiological hazards. Furthermore, the data from the storage test show another important result, namely the reproducibility of the antimicrobial efficacy in the in situ food model. Indeed, by using the natural antimicrobial agent at concentrations twice the MIC values, a clear and immediate inhibition of undesirable microorganisms was observed.

In addition, the use of medlar seed extract produces positive results on lipid oxidation.

Figure 5 shows the evolution of TBARSs, second-stage auto-oxidation products obtained by the oxidative cleavage of peroxides into ketones and aldehydes [90].

An increase in TBARSs during storage was detected in all samples. However, significant differences were found between the different batches. The most pronounced increase and consequently the highest values were found in the control samples. In contrast, the lowest TBARS values, reflecting lower oxidation, were found in the samples from the batches prepared with medlar seed extract. In an intermediate position were the samples prepared with the use of albedo. In accordance with the findings of other authors [91], plant extracts rich in polyphenols enable a containment of oxidative phenomena detectable with a lower TBARS content. Therefore, enrichment of fresh meat products with medlar seed extract is an effective solution to the oxidation problem and fulfills the clean-label qualification. The combination of the antimicrobial medlar seed extract with the fat replacer also solves the as-yet-unsolved problem [16] of the oxidation of fat replacers or fat mimetics.

#### 3.3.2. Anti-*Listeria* Validation

The effectiveness of anti-*Listeria* action was evaluated by performing a specific challenge test. Figure 6 shows trend of *Listeria* in burgers intentionally inoculated with a multi-strain cocktail of *Listeria* and fortified with the plant extract alone (EL) or in combination with albedo AEL, with albedo alone (AL), or without any addition (CL) as a control batch. Significant differences (*p* < 0.05) were found to be time-dependent and batch-dependent. Specifically, batches CL and AL showed an increase in *Listeria* levels of more than one logarithmic cycle after the first day of refrigerated storage and 2.5 (CL) and 2.0 (AL) logarithmic cycles. No significant differences were found between batches CL and AL, at least until the third day of storage. Significantly different behavior (*p* > 0.01) to that characterizing the above-mentioned batches was exhibited by the samples of batches EL and AL. These latter batches, using extract without or together with albedo, showed a significant decrease in *Listeria* levels. Specifically, a decrease of about one logarithmic cycle in *Listeria* levels was observed in both batches on the sixth day of storage.

Therefore, it appears that the anti-*Listeria* activity is clearly attributable to the presence of the medlar seed extract. The *Listeria* decay observed in the samples with the extract confirms the efficacy of the extract even in situ in the food model. This is of relevance and represents a solution to the challenges that, as reported above [85], are still open in the food industry. Also noteworthy is the extent of the inhibiting action expressed by the extract. Other authors [92,93] have investigated the anti-*Listeria* effect expressed by the extracts in situ and found only a slowing down of growth or a growth arrest of the unwanted strain. In our case, a decay even greater than that found by other authors [94,95] using bacteriocin-producing bacteria was found.

## 4. Conclusions

The present study, with the goal of achieving “better meat”, led to solutions addressing one of the most pressing challenges concerning the meat and meat product industry by developing strategies based on the use of plant-based ingredients.

The results show that albedo as a fat substitute provides important nutritional benefits, giving innovative burgers a better place in the sphere of health expectations. In addition, albedo overcomes the sensory limitations inherent in many fat replacers or mimetics, by providing levels of enjoyment for all sensory attributes comparable to or even better than the control. However, this study confirms that albedo, like other fiber-rich fat mimetics, does not provide satisfactory microbiological quality and underscores the need for an appropriate natural antimicrobial agent. Medlar seed plant extract, rich in phenolic substances, showed interesting antimicrobial and antioxidant activity both in vitro and in situ. The findings are of particular interest when considering that a discrepancy in efficacy between in vitro and in situ conditions is often found. In vitro, significant anti-*Pseudomonas* and anti-*Listeria* activity was demonstrated without causing adaptation or resistance by the target microorganisms. The in vitro investigations, also supported by predictive tools, enabled the selection of the most appropriate natural microbial agent. Based on the consistency between the in vitro and in situ results, it can be hypothesized that the greater the in vitro knowledge of the performance and efficacy of the natural antimicrobial agent, the greater the probability of in situ success in the food model. The combined in situ use of medlar seed extract, as a natural antimicrobial, and albedo, as a fat replacer, enables burger quality to be met in its broadest aspects. In fact, the combined use also allows doubts about quality and microbiological safety to be bridged. Finally, the body of knowledge gained through the present study constitutes an important building block for the development of meat products matching the microbiological safety and closer to the health needs expressed by the market and citizens, as bearers of interests in the broad concept of one health.

## Figures and Tables

**Figure 1 foods-13-03229-f001:**
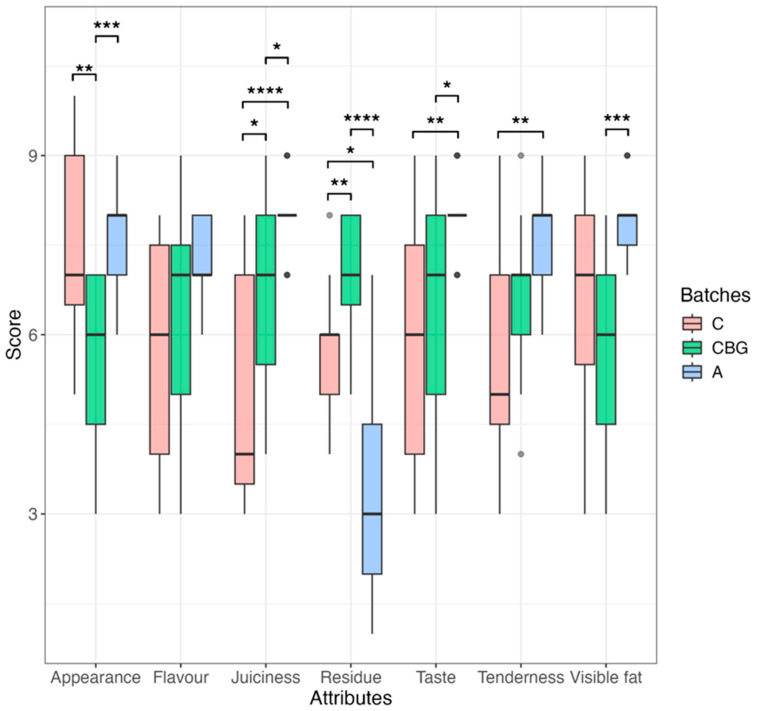
Box plot showing the acceptability level (9-point hedonic scale) of sensory attributes (appareance, flavour, juiciness, residue, taste, tenderness and visible fat) in samples from batch C (conventional burgers prepared with lean minced beef and beef fat), from the batch CBG (burgers prepared with the use of carob bean gum as fat replacer), and from batch A (burgers prepared with the use of albedo flour as fat replacer). Asterix indicates significant differences (*, *p* < 0.05; **, *p* < 0.01; ***, *p* < 0.001; ****, *p* < 0.0001) in attributes between the 3 batches. Statistical test was carried out using the Kruskal–Wallis test, followed by Dunn’s post hoc test for multiple comparison.

**Figure 2 foods-13-03229-f002:**
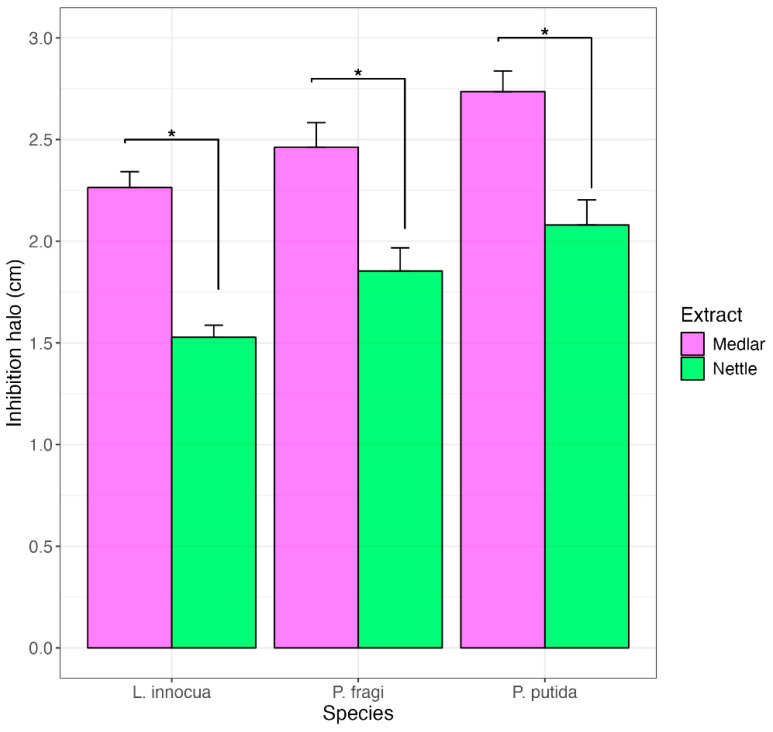
Bar plot showing the inhibition halo (cm) of medlar and nettle extracts against *L. innocua*, *P. fragi*, and *P. putida*. Asterix indicates significance difference (*p* < 0.05) between extracts on the base of the statistical *t*-test calculated on 5 independent replicates.

**Figure 3 foods-13-03229-f003:**
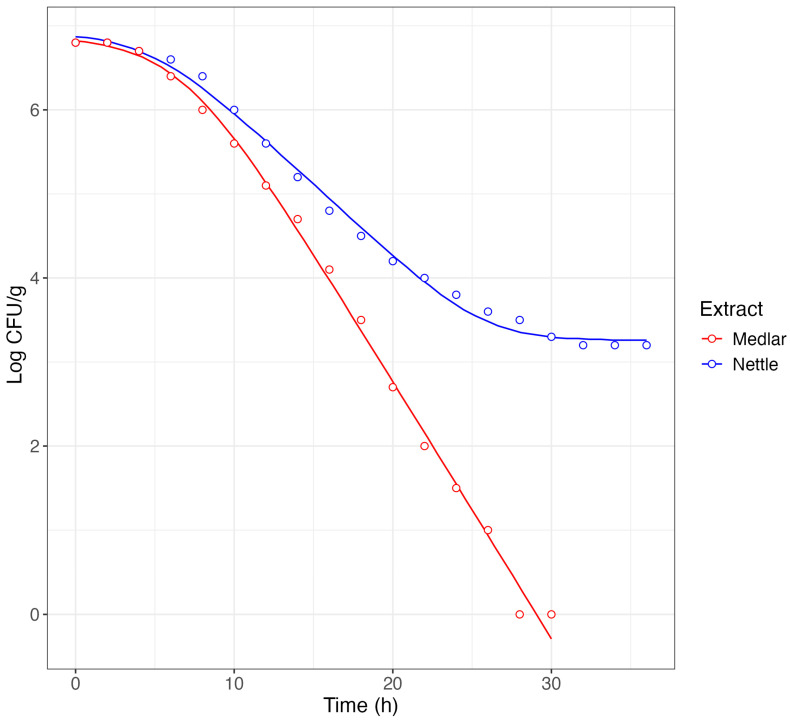
Model by Baranyi and Roberts [45] describing the inactivation of *L. innocua* treated with medlar seed extract and nettle extracts used in doses of 2 times the MIC values. Symbols represent the mean of experimental data obtained by three independent experiments and lines represent the model values.

**Figure 4 foods-13-03229-f004:**
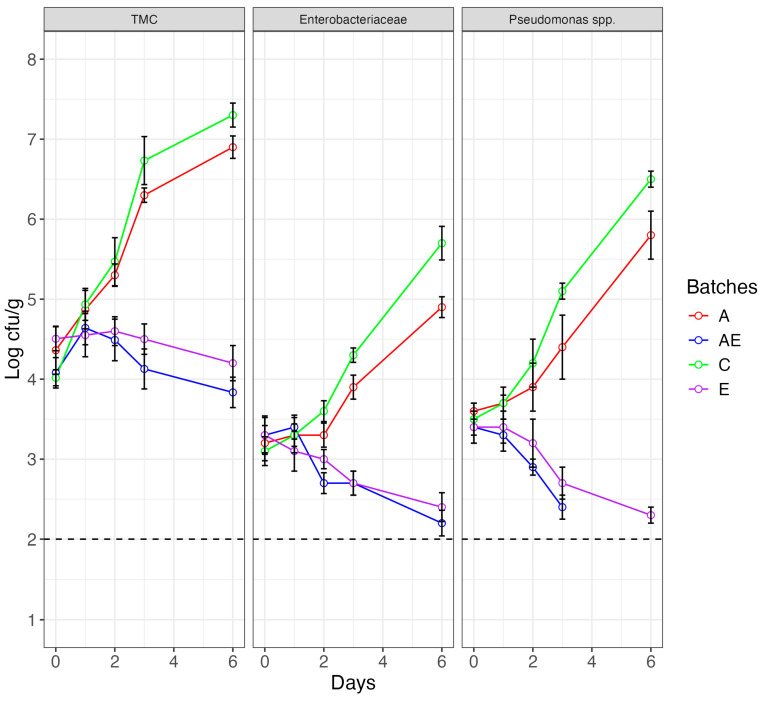
Line plots showing the evolution of TMC, *Enterobacteriaceae* and *Pseudomonas* spp. during the refrigerated storage period (6 days) in hamburgers prepared conventionally (batch C), with albedo (batch A), with medlar extract (batch E) and with the combination of albedo and medlar extract (batch AE).

**Figure 5 foods-13-03229-f005:**
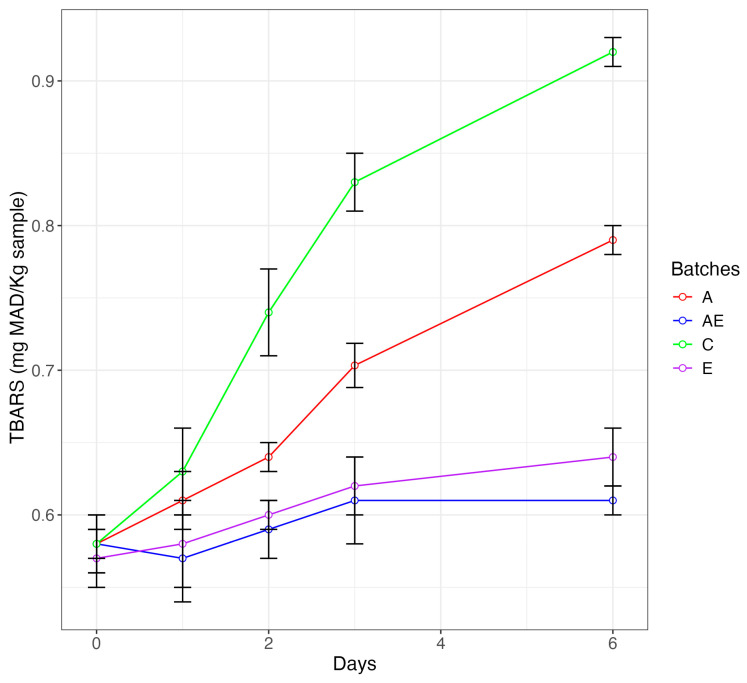
Line plots showing the thiobarbituric acid reactive substance (TBARS) values during the refrigerated storage in burgers from batches AE (containing 10% albedo (as fat replacer) and medlar extract (0.5 mL/100 g burgers)), A (containing the albedo (10%) as fat substitute), C (containing 10% beef fat (control batch)), and E (burgers containing medlar extract (0.5 mL/100 g burgers)).

**Figure 6 foods-13-03229-f006:**
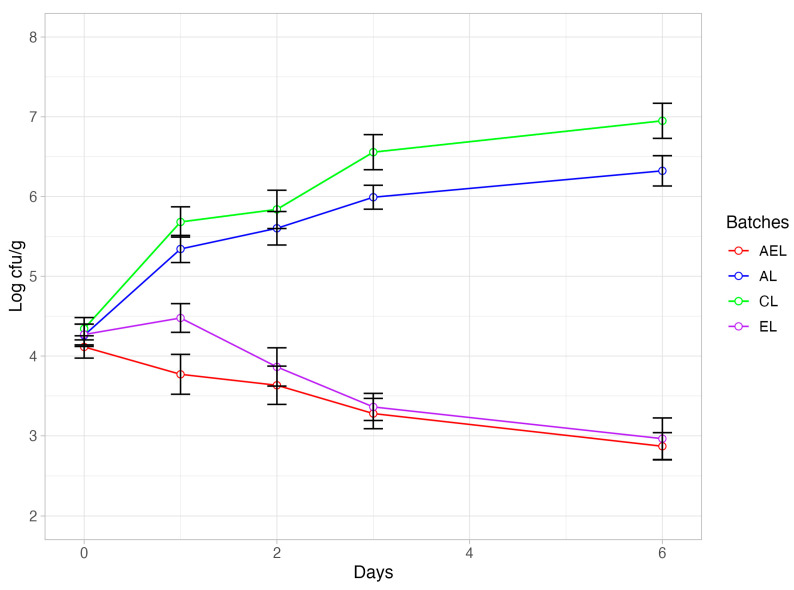
Line plots showing the challenge test of *Listeria* spp. during the storage of burgers from batch AEL, containing 10% albedo (as fat replacer) and medlar extract (0.5 mL/100 g burgers) and inoculated with multi-strain cocktail of *Listeria* spp.; batch AL, containing the albedo (10%) as fat substitute and inoculated with multi-strain cocktail of *Listeria*; batch CL, containing 10% beef fat (control batch) and inoculated with multi-strain cocktail of *Listeria*; and batch EL, burgers containing medlar extract (0.5 mL/100 g burgers) and inoculated with multi-strain cocktail of *Listeria.*

**Table 1 foods-13-03229-t001:** Chemical composition of hamburgers formulated conventionally (batch C) or with the addition of albedo (batch A).

Parametres	Batch C	Batch A
Moisture	65.54 ± 0.32 ^a^	69.52 ± 0.33 ^b^
Protein (d.m.)	59.95 ± 0.79 ^a^	59.29 ± 0.64 ^a^
Lipid (d.m.)	36.91 ± 0.41 ^a^	6.91 ± 0.33 ^b^
Ash (d.m.)	3.13 ± 0.21 ^a^	4.00 ± 0.17 ^b^
Dietary Fiber (d.m.)	n.d.	14.25 ± 0.41 ^b^
Carbohydrates (d.m.)	2.61 ± 0.26 ^a^	4.00 ± 0.23 ^b^
Kcal	203.26 ± 3.41 ^a^	118.95 ± 2.98 ^b^

Means within a row with different letters are significantly different (*p* < 0.05) on the base of statistical *t*-test. n.d., not detected.

**Table 2 foods-13-03229-t002:** Microbial load (Log CFU/g) of hamburgers immediately after preparation.

Microbial Groups	Batch C	Batch A
TMC	5.5 ± 0.2	5.4 ± 0.3
*Pseudomonas* spp.	4.8 ± 0.3	4.9 ± 0.1
*Enterobacteriaceae*	3.1 ± 0.2	3.2 ± 0.4
*Listeria* spp.	<100	<100

**Table 3 foods-13-03229-t003:** Polyphenol content and antioxidant activity of nettle leaf and medlar seed extracts.

Extracts	Polyphenols (mg GAE/g Dry Matter)	IC50 (mg/mL)
Nettle	125.64	21.53
Medlar	398.12	2.91

## Data Availability

The original contributions presented in the study are included in the article/Appendix A, further inquiries can be directed to the corresponding author/s.

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
