# Peer review of "Plant-Based Ingredients Utilized as Fat Replacers and Natural Antimicrobial Agents in Beef Burgers"

_foods, 2024, doi:10.3390/foods13203229_

Round 1

Reviewer 1 Report

Comments and Suggestions for Authors

The article presents the development of a hamburger with novel ingredients with emphasis on sensory, chemical composition and microbiological issues.

Following observations and comments to improve the work:

The title could be improved to make it more fluent, also the qualifiers high quality and innovative seem to be arbitrary and imaginary.

The structure of the abstract should be improved and should include in a balanced and clear way aspects about the importance of the topic or problem to be solved, methodology, main findings.

The scientific names of nettle leaves and medlar seeds should be included at least once in the document and then their common names can be used.

In l40 you can also include a sentence that talks about the impact of such products on human health.

What is the relationship between baking performance and hamburgers, it is not clear what they mean in l55.

It is important that the effects of new ingredients and additives on sensory and physical properties be addressed in a more in-depth and separate manner, and on the other hand, the issues of safety and microbiology L68-l61.

In l64, the relationship between moisture and microbiological issues is discussed, but no further comment is made on water activity in relation to the addition of fat replacers. 

In addition, examples could be given of the most common additives used to replace fat and improve microbiological stability and then discuss the ones used in the work.

The relationship between the clean label concept and the use of fat replacer and antimicrobial agents is not clearly presented.

The rationale for sentence l68-l70 is not clear. ....

The rationale and types of predictive microbiology assays used as criteria in l90 is not clear, and the somewhat convoluted wording from l80 to l92 needs to be revised.

In 2.1 the origin of the products in terms of processing and reproducibility is not clear. The process of obtaining Albedo is not indicated and for carob bean the information is ambiguous.

Regarding the preparations C, A, CF the wording is somewhat confusing in all cases they used 10% ? and it gives the impression that the formulations are not completely equivalent and therefore the comparison would not make sense....

It would be useful if the authors could present in a complete way the base formulation of the hamburgers.

On the sensory part, if the participants were not trained, how do they ensure the statistical significance of only 15 panelists, it is necessary that they justify and defend the decision in fundamental ways?

Regarding the cooking method, you could add a brief description.

The proximate analyses of l129 are not fully described, for example only some standard protein and fiber are indicated.

Only in l154 are the species used adequately identified with their scientific names. It would also be advisable to elaborate a little more on nettle and medlar in the introduction.

2.3.1 is very long and could be simplified.

2.3.2 Measuring instrument spectrophotometer?

In 2.4 the time period is not clearly mentioned.

Figure 1 would be better represented as a table with the same information.

If there are 4 batches, why do tables 1 and 2 present only two?

Table 3 talks about nettle and medlar in circumstances that for almost all the document talks about batch, it is necessary to review and standardize.

Evaluate if it is possible to use some mathematical model for the data in figure 3.

If possible, improve the resolution of Figures 4 and 5.

The discussion of the issues associated with storage test can be improved in terms of depth.

The conclusion should be improved to emphasize the knowledge generated, findings and possible applications.

Author Response

Thank you very much for taking the time to review this manuscript. Your comments were extraordinarily helpful and, in my opinion, helped to improve the quality and presentation of the article. Below are the detailed responses (Answer) and corresponding revisions. Revisions are shown in red in the text of the manuscript. 

Question (Q) 1. The title could be improved to make it more fluent, also the qualifiers high quality and innovative seem to be arbitrary and imaginary.

Answer (A) 1. The reviewer's observation is very helpful. The title has been revised by placing more emphasis on the main theme of the article. Therefore, the title has been changed to “Plant-based ingredients utilized as fat replacers and natural antimicrobial agents in beef burgers”

Q2.        The structure of the abstract should be improved and should include in a balanced and clear way aspects about the importance of the topic or problem to be solved, methodology, main findings.

A2.         The abstract was improved according to the reviewer's suggestions as follows:

This study aimed to find solutions based on the use of plant-based ingredients that would not only improve the nutritional quality of meat products but also ensure sensory and microbiological quality. Two fat-replacers, lemon albedo (Citrus lemon) and carob seed gum (Ceratonia siliqua), were studied for their effect on the nutritional and sensory quality of beef burgers by chemical analysis and panel testing. The antimicrobial activity of two plant extracts, from nettle (Urtica dioica) leaves and medlar (Eriobotrya japonica) seeds, was studied evaluating by plate test the intensity of inhibitory action and the minimum inhibitory concentration against Pseudomonas spp. and Listeria innocua strains. In addition, the antioxidant activity of both extracts were evaluated. Based on the results, lemon albedo and medlar seed extract were validated in a food model (beef burger) by a storage and a challenge test. The results of the storage test show that medlar seed extract prevents the formation of thiobarbituric acid reactive substances (TBARS) and ensures microbiological quality inhibiting Enterobacteriaceae and Pseudomonas spp. Anti-Listeria efficacy was confirmed in situ by challenge test results. In conclusion, fat replacers, even if ensuring nutritional and sensory quality, do not meet microbiological quality. The use of fat substitutes should be combined with the use of extracts selected for their antimicrobial activity.

Q3.        The scientific names of nettle leaves and medlar seeds should be included at least once in the document and then their common names can be used.

A3           The scientific names of nettle and medlar have been included in the introduction when they are mentioned for the first time (L90 and L103) and in the abstract.

Q4.        In l40 you can also include a sentence that talks about the impact of such products on human health.

A4.         A sentence on the impact of meat products on human health was inserted as suggested. From L42 to L44, the new sentence and the relevant literature reference have been included: Meat and meat products are the most controversial food products due to their disadvantageous position in terms of environmental sustainability, challenging climate change and promoting health and safety. 

Q5.        What is the relationship between baking performance and hamburgers, it is not clear what they mean in l55.

A5.         Guedes et al. (2016) and Selani et al. (2016) found that the use of fiber as a fat substitute in processed meat products improves performance during cooking. This concept was further explained in L59 as follows:

Several authors found an improved cooking yield and rheological characters in chicken patties and beef burgers when plant fibers were used [14-15].

Q6.        It is important that the effects of new ingredients and additives on sensory and physical properties be addressed in a more in-depth and separate manner, and on the other hand, the issues of safety and microbiology L68-L61.

A6.         The appreciated suggestions were accepted. In particular, the effects on sensorial and microbiological quality were examined in depth and separated into different sentences as reported below (L63-L76)

However, to date, although many fat substitutes are available on the market, there are still many challenges to be overcome to improve their sensory perception [16]. These authors recently reported that the reduction and replacement of fat in meat products leads to changes in physiological and sensory properties. They emphasize that unresolved questions remain concerning texture, hardness, oxidative stability, juiciness, viscosity and overall acceptability. More specifically, the Authors of the recent review [16], point out that each macro-typology of fat substitutes shows substantial limitations on the sensory level. Whey protein-based substitutes have negative effects on hardness, chew-ability and adhesiveness; carbohydrate-based substitutes (starch or inulin) improve hardness but have disadvantages such as graininess and burnt flavor after cooking; and finally, lipid-based substitutes (oils from vegetable sources) are characterized by oxidation problems and in some cases gumminess. In addition, the use of fat replacers as alternative ingredients does not always guarantee microbiological quality and safety. In this regard, it should be considered that the microbiological quality of fresh meat is known to be a critical concern since its intrinsic characteristics allow the growth of various spoilage and pathogenic microorganisms [17-18]. The addition of fat replacers, which further increase the moisture content of the meat product [19-20], can further affect the microbiological quality [21].

Q7.        In l64, the relationship between moisture and microbiological issues is discussed, but no further comment is made on water activity in relation to the addition of fat replacers.

A7.         The relationship was further discussed in depth (L78-L85)

…The latter authors reported that the use of albedo in fermented meat products caused an increase in moisture and water activity (aw). Consequently, the use of fat replacers should not be separated from the use of antimicrobial preservatives. However, the use of conventional preservatives would conflict with latest clean label qualification requirements that encouraging the reduction of additive use. Whereas, the use of natural additives would satisfy the two opposing requirements (microbiological safety and clean label). The antimicrobial activity of plant extracts and their protective action in food systems have been extensively studied in recent decades [22-23-24-25].

Q8.        In addition, examples could be given of the most common additives used to replace fat and improve microbiological stability and then discuss the ones used in the work.

A8. The reviewer's suggestion contributes to improving the quality of the paper and its discussion. Therefore, it has been accepted and examples of fat substitutes already investigated and used have been included since the introduction (L67-L73). Two sentences have been included as follows:

More specifically, the Authors of the recent review [16], point out that each macro-typology of fat substitutes shows substantial limitations on the sensory level. Whey protein-based substitutes have negative effects on hardness, chew-ability and adhesiveness; carbohydrate-based substitutes (starch or inulin) improve hardness but have disadvantages such as graininess and burnt flavor after cooking; and finally, lipid-based substitutes (oils from vegetable sources) are characterized by oxidation problems and in some cases gumminess.

Q9.        The relationship between the clean label concept and the use of fat replacer and antimicrobial agents is not clearly presented.

A9.         The relationship has been better clarified as follows (L80-L83)

Consequently, the use of fat replacers should not be separated from the use of antimicrobial preservatives. However, the use of conventional additives would conflict with clean label qualification requirements that encouraging the reduction of additive use. Whereas, the use of natural additives would satisfy the two opposing requirements (microbiological safety and clean label). The antimicrobial activity of plant extracts and their protective action in food systems have been extensively studied in recent decades.

Q10.     The rationale for sentence l68-l70 is not clear. ....

A10.      The sentence was rephrased after a lengthy discussion of the limits of fat substitutes and knowledge about antimicrobial plant extracts. A new sentence was inserted at line 93-96:

Although there is plenty of evidence of protection for extracts of leaves and, in some cases, also of plant seeds, to our knowledge there is a lack of in vitro or in food-model studies on the combined use of antimicrobial plant extracts and fat substitutes.

Q11.     The rationale and types of predictive microbiology assays used as criteria in l90 is not clear, and the somewhat convoluted wording from l80 to l92 needs to be revised.

A11.      The extract was chosen by investigating not only the minimum inhibitory concentration but also by applying predictive methods to understanding resistance and adaptation of target strains to the extract. Finally, the selected antimicrobial extract used in combination with fat replacers was validated in the food model by storage and challenge tests.

As suggested by the reviewer, the use of predictive models (based on the analysis of live data by means of the Baranyi and Roberts equation) aimed at investigating the behavior of the microorganisms under study with respect to the extract was justified and better clarified as follow:

The vegetal extract with protective activity was selected on the basis of its antioxidant activity and its anti-Pseudomonas and anti-Listeria capacity. To better understand the specific action of the extract against the micro-organisms in question, the microbial decay in the presence of the extracts was also studied through the application of microbiology predictive models aimed at ascertaining possible adaptations or resistance of the strains to the protective extracts. Finally, the selected antimicrobial extract and fat replacer were validated in the food model by means of storage and challenge tests.

Q12.     In 2.1 the origin of the products in terms of processing and reproducibility is not clear. The process of obtaining Albedo is not indicated and for carob bean the information is ambiguous.

A12.      The authors have taken up the reviewer's suggestions, and the processes for obtaining the two fat substitutes have been reported in more detail:

Albedo: obtained by freeze-drying the crude portion of albedo, a by-product of lemon (Citrus lemon) processing, as reported by Tremonte et al. Specifically, the fresh raw material was treated at 90°C for 5 minutes to restore it from any microorganisms, frozen at -30°C until use, then subjected to freeze-drying. To obtain a powder with a particle size of less than 0.417 mm, an appropriate grinder and sieves were used.

Carob bean gum (CBG): purchased from manufacturers (CioKarrua Ltd.) in Southern Italy, was obtained, as reported by the manufacturer, from hulled carob seeds, sieved and ground to obtain native carob seed gum which was, then, extracted with distilled water (90°C for 60 minutes) and precipitated in isopropanol. The white fibrous precipitate formed was collected by filtration, dried under vacuum and ground into a fine powder.

Q13.     Regarding the preparations C, A, CF the wording is somewhat confusing in all cases they used 10% ? and it gives the impression that the formulations are not completely equivalent and therefore the comparison would not make sense....

A13.      The description of the batches, as suggested by the reviewer, has been clarified. Paragraph 2.2 has been revised (L135-L142):

C: burgers prepared with the mixture of salted and minced lean meat (90%) added with beef fat (10%). Specifically, the fat was taken from non-low-melt cuts and shredded. The batch was used as a control.

A: burgers prepared with the mixture of salted and shredded lean meat (90%) added with Albedo (10%). Freeze-dried albedo was previously dissolved in water, gelled and shredded in the same way as fat.

CBG: burgers prepared from the mixture of salted and shredded lean meat (90%) added with carob bean gum (10%). The carob bean gum was suspended in water, gelled and shredded.

Q14. It would be useful if the authors could present in a complete way the base formulation of the hamburgers.

A14.      The base formulation has been described(L129-L134):

The same basic formulation consisting of lean beef, sodium chloride (2%) and ascorbic acid (0.1%) was used for all batches.

Specifically, the lean meat from the brisket cut, suitably defatted and deprived of visible connective tissue, was minced and added with sodium chloride and ascorbic acid. The basic formulation after mixing was divided into three aliquots corresponding to the three batches

Q15.     On the sensory part, if the participants were not trained, how do they ensure the statistical significance of only 15 panelists, it is necessary that they justify and defend the decision in fundamental ways?

A15.      I thank the reviewer for the opportunity to better describe the characteristics of the panel of judges who conducted the sensory analysis (section 2.2.1). The sensory evaluation was conducted by 15 panelists who were not officially qualified (i.e. not registered whit official organizations) but recruited from among food technology experts (researchers and PhD students of the Department of Agricultural, Environmental and Food Sciences - University of Molise, Campobasso, Italy) and specifically trained in the sensory analysis of meat products with particular reference to hamburgers. Specifically, the training period started with 45 potential judges and continued for 5 weeks with two weekly sessions, resulting in a group of 15 judges with a satisfactory uniformity of response. The group of judges met in three sessions and in each session evaluated three burgers each for one batch. Sensory analyses were then performed in triplicate.

Q16.     Regarding the cooking method, you could add a brief description.

A16.      The cooking methods were detailed: the cooking procedure was performed in a cooker according to the method described by the American Meat Science Association methodology (AMSA, 2015) [39]. Cooking was conducted until either a final internal temperature of 72 C was reached or recorded at the geometric center of each burger using a hypodermic probe thermocouple (model HYPOK60 ITSensor, Rovigo - Italy). (L156-158).

Q17.     The proximate analyses of l129 are not fully described, for example only some standard protein and fiber are indicated.

A17.      The description was clarified in the text (L165):

Protein, fat, carbohydrate, fiber, and ash and moisture, content were determined according to the official methods of AOAC 2023 (Protein content was determined using the Kjeldhal method; fat content in accordance with the AOAC method 960.39 using a Soxhlet apparatus; Dietary fiber content was determined using the AOAC method 985.29; carbohydrates were calculated by difference).

Q18.     Only in l154 are the species used adequately identified with their scientific names. It would also be advisable to elaborate a little more on nettle and medlar in the introduction.

A18.      Done

Q19.     2.3.1 is very long and could be simplified.

A19.      Done

Q20.     2.3.2 Measuring instrument spectrophotometer?

A20.      The description has been improved and the measuring instrument (spectrophotometer) has been reported (L236-237)… the absorbance was measured at 517 nm using a BioSpectrometer (Eppendorf, Hamburg, GE).

Q21.     In 2.4 the time period is not clearly mentioned.

A21.      Done

Q22.     Figure 1 would be better represented as a table with the same information.

A22.      Authors, accepting the reviewer's suggestions, the table is included as supplementary material

Q23.     If there are 4 batches, why do tables 1 and 2 present only two?

In order to choose the best fat replacer, three batchers were prepared: C, as control; A, containing albedo as fat replacer; and CBG, with carob bean gum as fat replacer. Since the sensory analysis showed a preference for the samples of batch A (with albedo) and a low acceptability of those of batch CBG, the data of the latter were not shown in the table. However, accepting the reviewer's observation, the data although not shown in the table was commented in the results section (L351-355): “…Chemical results show that both albedo and carob bean gum used as a fat-replacers significantly influenced the chemical and nutritional composition of the burgers with a relevant increase in fiber and a reduction in fat content. However, since carob seed flour did not meet with a good sensory acceptance, attention was turned to the effect produced by the use of albedo.”

Q24.     Table 3 talks about nettle and medlar in circumstances that for almost all the document talks about batch, it is necessary to review and standardize.

A24.      Table 3 has been aligned with Figure 2, both of which describe the characteristics of plant extracts with protective activity.

Q25.     Evaluate if it is possible to use some mathematical model for the data in figure 3.

A25.      The inactivation curve of L. innocua in the presence of nettle leaf extract and medlar seed was described by applying the model equation of Baranyi and Roberts (45). This information was included in the figure caption (L469) as well as in Materials and Methods (L230) and in the description of results (L449-452).

Q26.     If possible, improve the resolution of Figures 4 and 5.

A26.      DONE

Q27.     The discussion of the issues associated with storage test can be improved in terms of depth.

A27.      The discussion relating to the storage test has been revised, reporting the advantages and disadvantages attributable to the use of albedo and how the latter are overcome thanks to the use of medlar seed extract. Substantial revisions are shown in red by the line 506 to te line 536.

Q28.     The conclusion should be improved to emphasize the knowledge generated, findings and possible applications.

A28.      DONE

Reviewer 2 Report

Comments and Suggestions for Authors

The study focuses on combining fat replacers with antimicrobial plant extracts, a novel approach in line with consumer demand for healthier, cleaner-label products. The manuscript combines sensory, microbiological, and chemical analyses to evaluate the product’s quality and safety. The manuscript presents an interesting topic on meat product innovation, integrating fat-replacement strategies with natural antimicrobial solutions.

While extracts' antimicrobial and antioxidant properties are explored, the manuscript could provide more detail on why these specific extracts were chosen over others more commonly used in the industry. A brief comparison with alternative options would make this stronger.

Line 30 - The phrase "The results a enrich knowledge" seems awkward.

Introduction

Line 41 - "Less but better meat' is the pragmatic approach..." what did the authors mean with "better meat"? what particularly plant-based ingredients are needed?

Lines 62-65—The potential adverse effects of fat replacers increasing moisture content are briefly introduced. It would be helpful to expand on this to provide a more complete understanding of the risks associated with fat replacement.

Methodology

Lines 430-436 - The manuscript discusses the differences between in vivo and in vitro tests, but needs to provide a more in-depth explanation of these variations.

The microbiological analysis is thorough, but it would be helpful to discuss how these results translate into real-world shelf-life implications.

Line 106-107 - fat replacers. Why were these chosen over other common fat replacers?

Line 120-127—The sensory analysis section lacks detail on the panelists’ training level.

Results

Line 273-283 - The sensory analysis results for albedo and carob bean gum could benefit from additional statistical information to better interpret the differences in sensory attributes.

Line 330-338—The microbial load in the hamburgers is discussed but not clearly linked to potential implications for shelf life or safety. A deeper analysis, perhaps including water activity or other preservation-related factors, would improve this section.

Conclusions

The study demonstrates that seed extracts are highly effective antimicrobial agents. The potential benefits of fat replacers, such as albedo, are evident, but the long-term effects on product stability and consumer acceptance have not been fully explored.

Line 513-534: Comments on plant-based fat replacers should be more directly related to current consumer trends toward healthier meat alternatives.

Author Response

Thank you very much for taking the time to review this manuscript. Your comments were extraordinarily helpful and, in my opinion, helped to improve the quality and presentation of the article. Below are the detailed responses (Answer) and corresponding revisions. Revisions are shown in red in the text of the manuscript. 

Cemment1: The study focuses on combining fat replacers with antimicrobial plant extracts, a novel approach in line with consumer demand for healthier, cleaner-label products. The manuscript combines sensory, microbiological, and chemical analyses to evaluate the product’s quality and safety. The manuscript presents an interesting topic on meat product innovation, integrating fat-replacement strategies with natural antimicrobial solutions.

While extracts' antimicrobial and antioxidant properties are explored, the manuscript could provide more detail on why these specific extracts were chosen over others more commonly used in the industry. A brief comparison with alternative options would make this stronger.

Replay: The reviewer's comments offered very interesting review insights that, in my opinion, allowed for an improvement in the presentation of the study conducted and the results obtained.

Question (Q) 1.              Line 30 - The phrase "The results a enrich knowledge" seems awkward.

Answer (A) 1.  The sentence has been modified by explicating what are the main results produced by the study

Introduction

Q2.        Line 41 - "Less but better meat' is the pragmatic approach..." what did the authors mean with "better meat"? what particularly plant-based ingredients are needed?

A2.         The description of the “less but better meat” concept has been better defined by reparafrasing the sentences from line 46 to line 49.

The strategies that to achieve better meat are diverse, such as the reduction of fats or their replacement by alternative ingredients (plant-based fat substitute or plant-based fat mimetic), the reduction of synthetic additives (using natural antimicrobial agents) and increasingly less invasive ways of preparation and consumption, are just some of the issues that are driving the challenge for a better meat

Q3.        Lines 62-65—The potential adverse effects of fat replacers increasing moisture content are briefly introduced. It would be helpful to expand on this to provide a more complete understanding of the risks associated with fat replacement.

A3.         As suggested by the reviewer, the concept has been further detailed and clarified as follows (L77-L85): The addition of fat replacers, which further increase the moisture content of the meat product [19-20], can further affect the microbiological quality [21]. The latter Authors reported that the use of albedo in fermented meat products caused an increase in moisture and water activity (aw). Consequently, the use of fat replacers should not be separated from the use of antimicrobial preservatives. However, the use of conventional preservatives would conflict with the latest clean label qualification requirements. Whereas, the use of natural additives would satisfy the two opposing requirements (microbiological safety and clean label). The antimicrobial activity of plant extracts and their protective action in food systems have been extensively studied in recent decades [22-23-24-25].

Methodology

Q4.        Lines 430-436 - The manuscript discusses the differences between in vivo and in vitro tests, but needs to provide a more in-depth explanation of these variations.

A4.         Taking into account the reviewer's suggestion, the discussion on the question of discrepancy (found by other Authors) between in vitro and in vivo results was better clarified in Section 3.4. Since this discrepancy (according to the bibliography) is still an emerging challenge, in our study the antimicrobial efficacy of natural extracts was validated through tests conducted in food models by storage (3.4.1) and challenge tests (3.4.2). The results from the in vivo tests did not reveal any discrepancies with what was observed in vitro. This is precisely an important result that is also highlighted in the conclusions.

The microbiological analysis is thorough, but it would be helpful to discuss how these results translate into real-world shelf-life implications.

Q5.        Line 106-107 - fat replacers. Why were these chosen over other common fat replacers?

A5.         The two fat substitutes (albedo and carob bean gum) were chosen for their intrinsic features, being rich in fiber, and for their sustainability, being obtained from citrus processing by-products or resilient plants. This consideration was added in the text “…chosen for their sustainability and high fiber content..” (L115)

Q6.        Line 120-127—The sensory analysis section lacks detail on the panelists’ training level.

A6.         I thank the reviewer for the opportunity to better describe the characteristics of the panel of judges who conducted the sensory analysis. The sensory evaluation was conducted by 15 panelists who were not officially qualified (i.e. not enrolled in official organisations) but recruited from among food technology experts (researchers and PhD students of the Department of Agricultural, Environmental and Food Sciences - University of Molise, Campobasso, Italy) and specifically trained in the sensory analysis of meat products with particular reference to hamburgers. Specifically, the training period started with 45 potential judges and continued for 5 weeks with two weekly sessions, resulting in a group of 15 judges with a satisfactory uniformity of response. The group of judges met in three sessions and in each session evaluated three burgers each for one batch. Sensory analyses were then performed in triplicate.

Results

Q7.        Line 273-283 - The sensory analysis results for albedo and carob bean gum could benefit from additional statistical information to better interpret the differences in sensory attributes.

As suggested, additional statistical information was included in the results (L319-L325) and Figure1 caption.

Q8.        Line 330-338—The microbial load in the hamburgers is discussed but not clearly linked to potential implications for shelf life or safety. A deeper analysis, perhaps including water activity or other preservation-related factors, would improve this section.

A8.         The reviewer's helpful suggestion was accepted by including a more in-depth comment regarding safety and implications on shelf-life in light of the higher aw and sugar content that characterizes samples obtained using albedo. Two new sentences have been added (L376-378); (L386-389):

“…However, the higher moisture and carbohydrate content due to the addition of albedo raises questions about the possible effect on the naturally contaminating microbiota. … Considering also that burgers obtained by the addition of albedo are also characterized by slightly higher water activity (aw 0.969) values than those of the control (aw 0.961), the need for a protective strategy to safeguard microbiological safety and adequate shelf-life becomes evident.”

Conclusions

The study demonstrates that seed extracts are highly effective antimicrobial agents. The potential benefits of fat replacers, such as albedo, are evident, but the long-term effects on product stability and consumer acceptance have not been fully explored.

Q9.        Line 513-534: Comments on plant-based fat replacers should be more directly related to current consumer trends toward healthier meat alternatives.

A9.         DONE